# The Equity of Basic Educational Facilities from the Perspective of Space

**Qiya Huang** [1,2,*], **Xijuan Cui** [3] **and Libang Ma** [3]

1   School of Psychology, Northwest Normal University, Lanzhou 730070, China
2   School of Education, Lanzhou University of Arts and Science, Lanzhou 730030, China
3   College of Geography and Environmental Science, Northwest Normal University, Lanzhou 730070, China; 2020212699@nwnu.edu.cn (X.C.); malb0613@nwnu.edu.cn (L.M.)
*   Correspondence: 2021104116@nwnu.edu.cn

**Abstract:** Basic education pursues "balance" and "quality" under the premise of "universalization". High-quality and balanced education is the general strategy of international education. We used urban network tools to measure the spatial equity of three types of basic educational facilities (kindergarten, primary school, and middle school) in the main urban area of Lanzhou City, China, from the perspective of supply and demand. This can optimize the allocation of educational facilities, and make up for the shortage of basic educational facilities. It can also provide a scientific reference and new ideas for research on public service facilities. The conclusions are: (1) The distribution of basic educational facilities presented a typical pattern of belt-shaped clusters, river trends, and dense east and sparse west. The spatial dislocation between facilities and residential buildings was significant and occurred in numerous instances. (2) The supply of basic educational facilities was weak. There were significant differences in spatial accessibility among different types and regions. The spatial accessibility of kindergartens (34.83%) was the best with regard to walking conditions, followed by primary schools (27.43%) and middle schools (21.11%). (3) The distribution of basic educational facilities was affected by factors such as historical development, natural geography, social economies, and the travel behaviors of residents; (4) It is necessary to improve the problem of spatial imbalance through the implementation of refined planning management and resource allocation of infrastructure, the construction of a "community life circle", and the establishment of an early warning mechanism for academic degree attainment combined with big data.

**Keywords:** basic educational facilities; spatial equity; accessibility; optimal allocation; Lanzhou City; China

## 1. Introduction

With the advent of the era of the knowledge economy, education is becoming more and more important in social development. The spatial distribution and allocation of educational resources are very important to educational equity and have far-reaching significance for the realization of the sustainable development of society [1]. As one of the important indicators to measure social equity, educational equity is related to the vital interests of residents [2]. It is necessary to take into account the overall development of society and the needs of individuals in the process of allocating educational resources. Educational resources are the decisive factor for a country's level of development and international competitiveness. The unbalanced allocation of educational resources has become a prominent problem in the development of education, as shown in the spatial mismatch of educational resources [3]. The urban basic educational facility is an important factor in promoting educational equity. Its accessibility and equity are closely related to social livelihood, social equity, and the overall situation of new urbanization. This has always been a research hotspot in geography, education, and other disciplines.

The spatial allocation of educational resources is the focus of research on accessibility and equity. At the macro level, it is the allocation of resources among regions of different scales [4]. At the micro level, schools are the basic unit of resource allocation. The differences in resource allocation between regions are jointly determined by the distribution of schools and resource allocation [5]. The adjustment of the distribution of educational resources is an important way to achieve goals from the stage of "popularization of basic education" to that of "balanced allocation of resources". There are many existing studies on basic education from the perspective of sociology, economics, political science, psychology, etc. However, there are few studies on the distribution of educational resources, standards, characteristics, mechanisms, and laws from a spatial perspective. Space is one of the basic dimensions of the existence of things. The key issues in basic education research, such as educational equity and balance and the allocation of educational resources, are closely related to the distribution of educational resources. Therefore, we should pay attention to the distribution of education. In addition, spatial information mining provides a new perspective for research regarding the layout of basic educational resources.

Early studies of Western scholars on the spatial allocation of basic educational resources and the layout of educational facilities were guided by the location theory of public facilities. By constructing an econometric model, scholars took the spatial balance of efficiency and equity as the goal [6,7]. They analyzed the relationship between education policy and the spatial layout of basic educational resources [8], and measured residents' satisfaction with the spatial equity of basic educational resources [9,10]. As the object of concern expands from students to government education departments, schools, teachers, and other stakeholder groups, the requirements for school layouts expand from a single goal of accessibility to multiple factors, such as the school's scale benefit characteristics and education input and output relationship. The research method has changed from the optimal model to the multi-criteria decision-making model (MCDM) [11]. In addition, some scholars have studied the relationship between education policy, residential differentiation, and the distribution of educational resources by introducing the research theories and methods of political economy, urban sociology, and educational economics [12,13]. With the development of GIS technology, scholars have paid more attention to the spatial allocation of basic educational resources, including school district division [14,15], agglomeration mode [16], accessibility measurement [17,18], spatial location [19], etc. Regarding education, accessibility studies have analyzed preschools [20], and have considered primary or elementary [21,22] or secondary or high schools [23,24], or both primary and secondary schools [25]. Some studies included all educational facilities in the area covered by the research [26,27], but without evaluating the differences in quality between schools. Studies have shown that school choices and the transportation modes for commuting can differ according to school type [28]. The research on the distribution of basic educational resources in China started in the 1990s. Scholars have tried to study the topic from the perspectives of geography, education, economics, and society, resulting in a large number of multidisciplinary studies. These studies have been based on real problems and have developed rapidly. The research has mainly focused on the spatial distribution of basic educational resources and its influencing factors [29,30], the spatial accessibility and layout optimization of basic educational resources [31,32], the equalization of basic education public services and its influencing factors [33], and the relationship between basic education and population agglomeration [34].

To sum up, from the perspective of research, existing studies based on the perspectives of pedagogy, economics, sociology, and psychology are systematic. The research on educational facilities based on space has mainly focused on planning and geography, including the layout mode of facilities, configuration standards, institutional mechanisms, etc. [35]. There are few studies on the supply and accessibility of educational facilities [36]. In terms of the spatial scale of the research, most studies focused on the level of administrative divisions [10]. There are few small-scale studies on the optimization of the layout of educational resources at the street level [16]. From the perspective of research methods, scholars used a

single model to quantitatively evaluate the layout optimization of regional public service facilities. Therefore, the evaluation results are one-sided. There is a need for a scientific and systematic means to improve the level of education service supply from the perspective of the equalization of basic educational facilities and services. This paper took the main urban area of Lanzhou City, Gansu Province, China as the case area, took the basic educational facilities as the research object, and took residential buildings as the smallest research scale. We analyzed and evaluated the spatial distribution of supply, demand, and the accessibility of basic educational facilities for each street through urban network analysis tools. This provides a strategy for the equitable allocation of space for various types of educational facilities for each street.

## 2. Data Sources and Research Methods

### 2.1. Overview of the Study Area

Lanzhou is located in northwestern China, and in the central part of Gansu Province. It is located at 102°36′–104°35′ east longitude and 35°34′–37°00′ north latitude. It is next to Wuwei City and Baiyin City in the north, Dingxi City in the east, and Linxia Hui Autonomous Prefecture in the south, with a total area of $1.31 \times 10^4$ km$^2$ and an urban area of 1631.6 km$^2$. The terrain of Lanzhou is high in the west and south, and low in the northeast. The Yellow River flows from southwest to northeast, crosses the whole area, cuts through the mountains, and forms a bead-shaped valley interspersed with valleys and basins.

Lanzhou City, the capital of Gansu Province, is located in the geometric center of China's land territory. It is an important central city, industrial base, and comprehensive transportation hub in northwest China, and a core node city of the Silk Road Economic Belt. Lanzhou has jurisdiction over Chengguan, Qilihe, Xigu, Anning, and Honggu districts and Yongdeng, Yuzhong, and Gaolan counties. We took four urban areas in Lanzhou as examples: Chengguan District, Qilihe District, Anning District, and the Xigu area, 103.58–103.94 east longitude, 36.02–36.15 north latitude. The total area was 213.19 km$^2$, and the total population was 4.36 million in 2021. There are 50 streets in the main urban area, and 236 communities (Figure 1). According to the Lanzhou Statistical Annual Report, in 2021, there were 164,700 students in primary school, 68,400 students in middle school, and 3275 students in kindergarten. The land areas and building areas of schools in the main urban area of Lanzhou City generally did not meet standards, and most schools have no room for development. The average school area per student and sport area per student were seriously insufficient. The average school area per student was less than 10 m$^2$ in the middle schools, and was less than 7 m$^2$ in the primary schools. The average sport area per student was less than 3 m$^2$ in both primary schools and middle schools, which were far below the national standard (24 m$^2$).

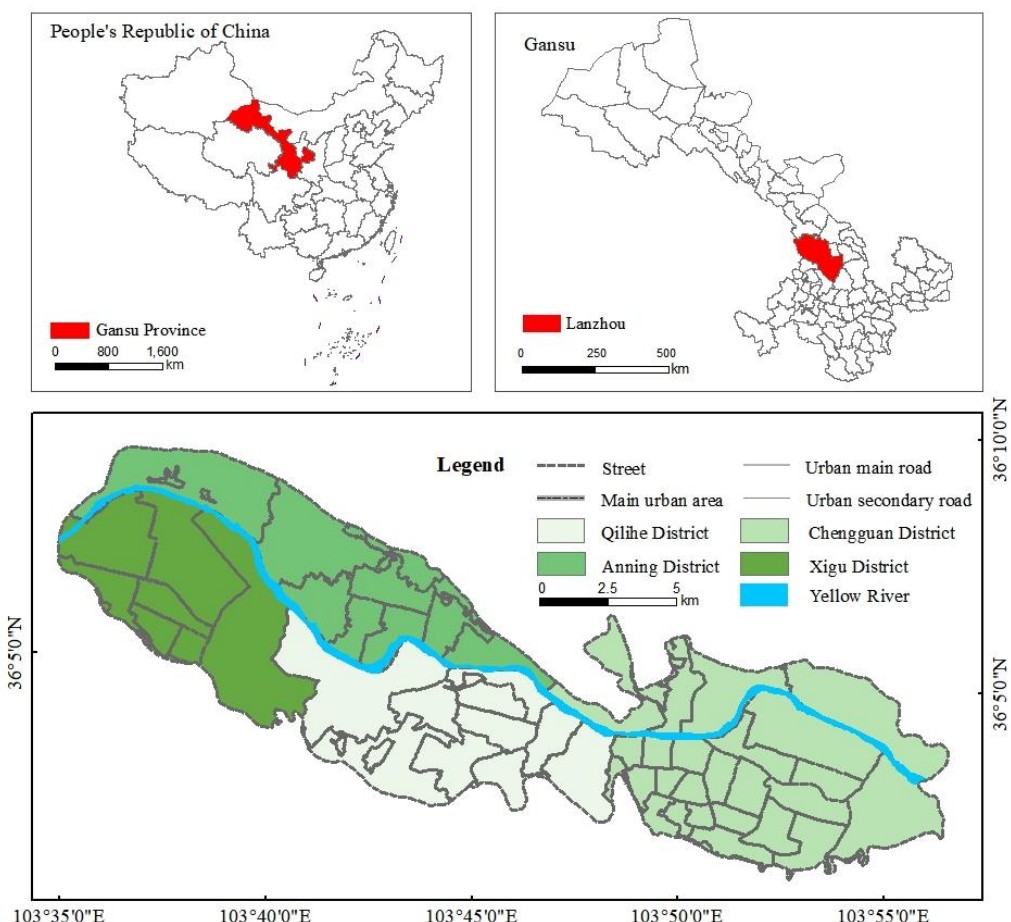

**Figure 1.** Study area map.

### 2.2. Data Sources and Data Processing

In this paper, middle schools, primary schools, and kindergartens in the urban area of Lanzhou represent urban basic educational resources. There are four main sources of data: (1) Data on basic educational facilities: kindergartens, primary schools, and middle schools in the urban area of Lanzhou City. Using Python crawlers deployed through the NasdaqAMAP map API, we first obtained 5287 pieces of scientific, educational, and cultural POI data, including names, addresses, latitudes and longitudes, and category codes. We used ArcGIS10.4 for spatial matching and spatial projection. According to the school list on the Lanzhou Education Website (www.edu.lanzhou.gov.cn), we finally identified 443 kindergartens, 162 primary schools, and 118 middle schools in the urban area of Lanzhou City, with a total of 723 basic educational facilities. (2) Outline of residential buildings: we used the Rivermap API to obtain building outline data for Lanzhou City from 2022. Then, we combined Google Earth and Baidu Street View to conduct manual interpretation and modification, and finally extracted the outlines of residential buildings in the urban area of Lanzhou City, with a total of 22,100 buildings. (3) Road data: we retrieved data from the open street map and retained the roads within 1000 m outside the urban area, and completed the road topology inspection. (4) The basic map: the administrative boundary of Lanzhou City (1:250,000) was from the geospatial data cloud (http://www.gscloud.cn/, accessed on 23 December 2022.).

### 2.3. Research Methods

2.3.1. Kernel Density Estimation

Rosenblatt and Emanuel Parzen first proposed Kernel Density Estimation to estimate the density of a dot or line pattern with the aid of a moving cell. For sample points

$\chi_1, \chi_1, \ldots \chi_n$, they used kernel estimation to find out the detailed distribution of attribute variable data. For two-dimensional data, the value of *d* is 2, and the kernel density estimation function is:

$$f_n(\chi) = \frac{1}{nh^2\Pi} \sum_{i=1}^{n} K\left[\left(1 - \frac{(\chi - \chi_i)^2 + (y - y_i)^2}{h^2}\right)\right]^2$$

where *K* is the kernel function; $(\chi - \chi_i)^2 + (y - y_i)^2$ is the distance from point $(\chi_i, y_i)$ to point $(x, y)$; *h* is the bandwidth; *n* is the number of points in the range; *d* is the dimensionality of the data.

### 2.3.2. Urban Network Analysis (UNA)

We conducted a supply-side analysis of basic educational facilities through the Urban Network Analysis (UNA) toolkit developed by the City Form Lab jointly established by the Massachusetts Institute of Technology (MIT) and the Singapore University of Technology and Design (SUTD). UNA is a tool for urban spatial network analysis that is not based on space syntax. The basic analysis idea is network + node, that is, roads are used as networks, and buildings and facilities are used as nodes. Nodes can be assigned weights, and geometric distances can also be measured. The UNA tool has two advantages over other research tools for the analysis of travel. First, it takes each residential building patch as a starting point, rather than the geometric center of the community. Therefore, the public service facilities that can be obtained by traveling 500 m, or another certain distance, are not always the same for residents in the same community. Second, the travel range of residents is not divided by the simple distance of a radius, but by the distance traveled on the road network. This can more realistically reflect the actual travel range of urban residents. Most existing studies performed with UNA tools were based on the ArcGIS platform. We found that the running speed of the ArcGIS platform was slower than that of the Rhino platform. In addition, the start and end points of the ArcGIS platform can only be used in one shapefile. This is not convenient for analysis. Therefore, we chose the Reach function in the Rhino platform, with the UNA tool for calculation.

The Reach function for the number of destinations reachable from a starting point with the shortest path distance is:

$$R[i]^r = \sum_{j \in G - \{I\}, d[i,j] \leq r} W[j] \tag{1}$$

where $R[i]^r$ represents the number of destination points *j* that the starting point *i* can reach within the buffer distance *r*; *G* is the traffic network; $d[i, j]$ represents the shortest distance between the starting point *i* and the destination point *j* in the network; $W[j]$ is the weight of the destination point *j*.

## 3. Results Analysis

### 3.1. Spatial Characteristics of Basic Educational Facilities

#### 3.1.1. The Distribution of Basic Educational Facilities

The spatial distribution of basic educational facilities in the urban area of Lanzhou City shows a certain degree of spatial agglomeration, with a typical spatial distribution pattern of bands and clusters (Figure 2a). Kindergartens were small in size and numerous. The number of kindergartens in Chengguan District and Qilihe District were 216 and 113, respectively, accounting for 74.25% of the total number. The number of kindergartens in Xigu District and Anning District were 55 and 59, respectively. The number of primary schools was small, with a total of 162 in the region. Chengguan District had the largest number of primary schools, reaching 78, accounting for 48.15% of the total number, followed by Qilihe District, with 37 primary schools. The number of primary schools in Anning District was the smallest, which was 23. The total number of middle schools was the

least, with only 118. The number of middle schools in Chengguan District was the largest, reaching 57, accounting for 48.31% of the total number, followed by Xigu District, reaching 29, and Anning District was the smallest, with only 13.

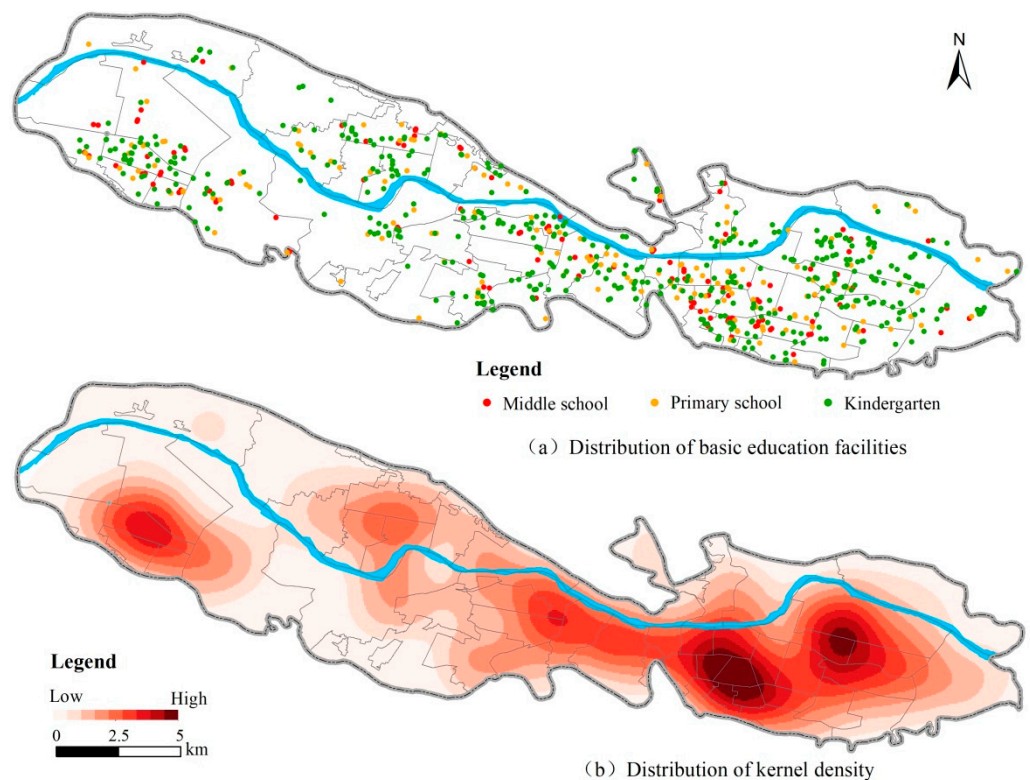

**Figure 2.** Distribution of basic educational facilities.

In Figure 2b, it can be seen that the spatial distribution density of basic educational facilities in the main urban area of Lanzhou City was in the shape of a strip along the Yellow River, dense in the east and sparse in the west. Yannan Road Street, Gaolan Road Street, and Jiuquan Road in the southeast formed two significant high-density agglomeration centers. Two secondary agglomeration centers were formed with Xigucheng Street and Jianlan Road Street. A low-agglomeration center was formed at Kongjiaya Street. In addition, the distributions of basic educational facilities in some areas were too sparse. The history of the rise and development of urban areas has significantly affected the distribution of basic education. Chengguan District, as an early developed urban area, has many basic educational facilities, which are evenly distributed. As an industrial development base, Xigu District has developed basic education for the education of children of employed workers.

### 3.1.2. Spatial Distribution of Residential Buildings

As the carrier of residents' daily life, the city is the direct demander of basic educational facilities, and it also affects the spatial distribution of basic educational facilities. However, urban populations are large in size, with diverse needs and behaviors. The daily life circle based on the individual not only fails to delineate the living circle, but also fails to solve the problem of the allocation of basic educational facilities. Therefore, the individual human must be transformed into the collective human. The actual carriers of population distribution are various types of residential areas. Therefore, scattered and individualized people can be integrated into groups of people at a certain scale based on the residential area (community, residential area, POI data). Based on the population data of the Seventh Census of Lanzhou City and the distribution data of urban residential buildings, we calculated the Pearson correlation between them. The correlation coefficient was 0.854, at

P< 0.01. Therefore, the distribution of urban residential buildings in Lanzhou can indicate the distribution of an urban population.

The spatial distribution of residential buildings and basic educational facilities determines the types of facilities and the number of facilities that residents can reach by traveling. Therefore, we analyzed the spatial distribution of the residential buildings based on the overall distribution and agglomeration of the facilities. Figure 3a shows that, in reference to the number of buildings in each district, Sijiqing Street, Xiuchuan Street, Chenping Street, Xiyuan Street, Yanbei Street, and Donggang Street each have a large number of buildings. The size of the district affects the number of buildings. As shown in Figure 3b, from the perspective of the agglomeration of buildings, the residential buildings in the narrow transitional area from east to west in Lanzhou City formed a high concentration center, which spread outward in circles. The western and central regions formed two secondary agglomeration centers. In addition, multiple small agglomeration centers were formed inside the region. From the kernel density of the spatial distribution of basic educational and residential buildings, the distribution of the two had a certain degree of overlap in the macro scale. However, there were large differences between the two in the micro scale, and the "dislocation" phenomenon between the two was significant. The equitable distribution of high-quality resources is the goal of narrowing the gap between urban and rural areas, and of social development, and the equity of access to basic educational resources is a key premise of the overall goal of achieving equity and efficiency of access to high-quality resources.

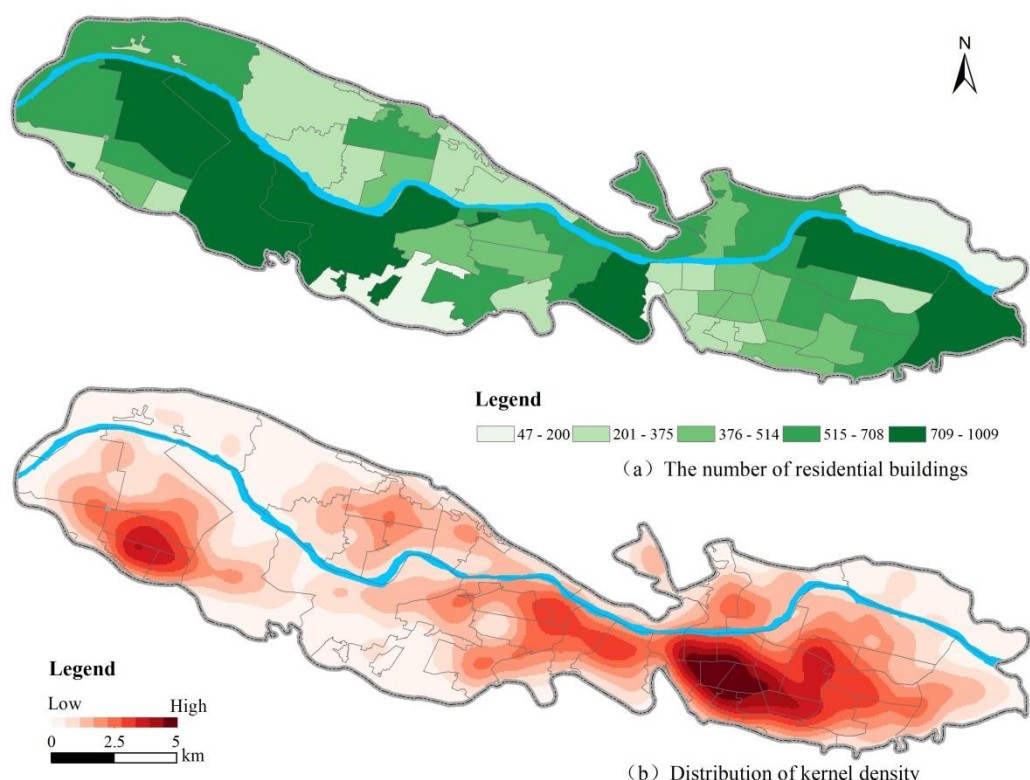

**Figure 3.** Spatial distribution of residential buildings.

## 3.2. Supply of Basic Educational Facilities

The balanced development of basic educational resources includes policy norms, indicators and mechanisms, and spatial equity. Therefore, we should pay attention to the differences between urban and rural areas, and the differences between regions. In implementing the development concept of "innovation, coordination, greenness, openness and sharing", China should better stimulate the vitality of urban and rural communities and put forward the concept of the community life circle. In the allocation of basic guaranteed

service elements in the urban community life circle, kindergartens, primary schools, and middle schools are the basic service elements of basic education. The service radii are 300 m, 500 m, and 800 m, respectively.

In this paper, the supply capacity of a basic educational facility is the number of residential buildings that it can serve. The larger the value, the greater the supply capacity. In Figure 4, it can be seen that there were great differences in the supply capacities of basic educational facilities in Lanzhou, and the spatial distribution was uneven. The overall supply capacity of the west of the city was higher than that of the east, and that of the north was higher than that of the south. In addition, there was a certain overlap in the supply capacity of kindergartens, primary schools, and middle schools. This indicates that the configuration of basic educational facilities in a certain region is often combined and matched with a certain continuity in terms of levels of education.

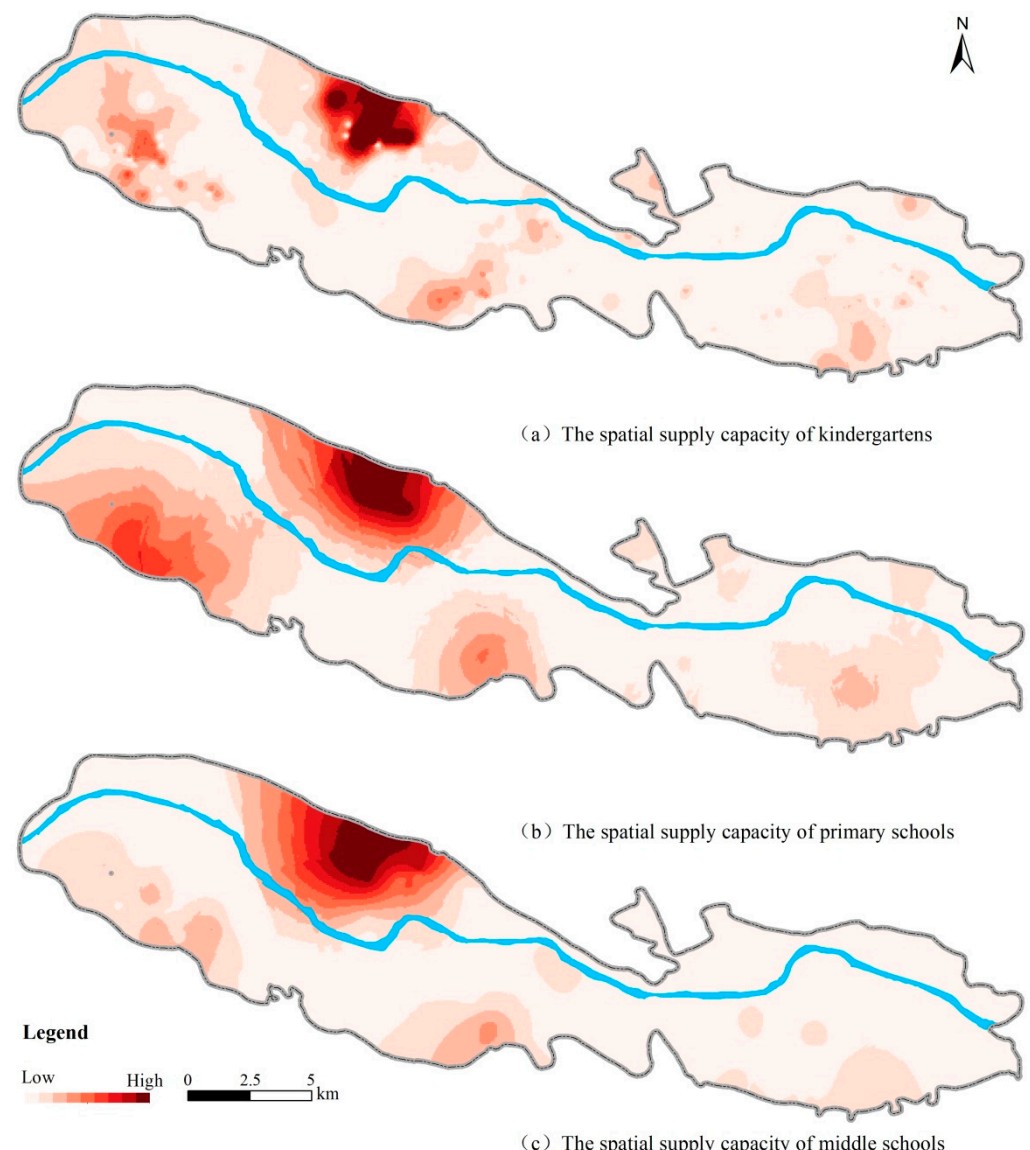

（a）The spatial supply capacity of kindergartens

（b）The spatial supply capacity of primary schools

**Legend**

Low　High　0　2.5　5 km

（c）The spatial supply capacity of middle schools

**Figure 4.** The spatial supply capacity of basic educational facilities.

If the existing services were in the range of foot travel, the total supply capacity of basic educational facilities, in descending order, was: kindergartens, primary schools, and middle schools. The supply capacity of kindergartens in Anning District was the highest at 71, followed by Xigu District at 67, and the lowest was in Chengguan District at 25. The internal differences in Anning District were the largest, with the highest supply of

381 and the smallest supply of 4. Qilihe District had the smallest differences in internal supply capacity, and the overall supply capacities of the internal streets were almost the same. The supply capacities of Jiaojiawan Street and Jingyuan Road in Chengguan District were higher than that of other streets. The supply capacity of primary schools in Anning District was higher than that of Xigu District, and the supply capacity of primary schools in Qilihe District was higher than that of Chengguan District. There were no significant differences in the supply capacity among the inner streets of Chengguan District. The supply capacity of Anning District's West Road Street was significantly higher than that of other streets, thereby improving the overall supply capacity. The supply capacity of each street in Xigu District varied greatly. The supply capacity of Gongjiawan Street in Qilihe District was significantly higher than that of other streets. The differences among the streets in Chengguan District were small. The supply capacity of middle schools was highest in Xigu District, with an average number of 54, followed by Anning District with 45, Qilihe District with 27, and Chengguan District with 19. As shown in Figure 5, there were significant internal differences among the streets of both Qilihe District and Anning District. Although the supply capacity of the streets in Chengguan District and Xigu District fluctuated, the overall changes were not significant. This is directly related to the number of types of basic educational facilities. In addition, since each region has its own hotspots of economic development, this can affect the number and distribution of residential buildings in the region. It can also affect the supply capacity of basic educational facilities, to a certain extent.

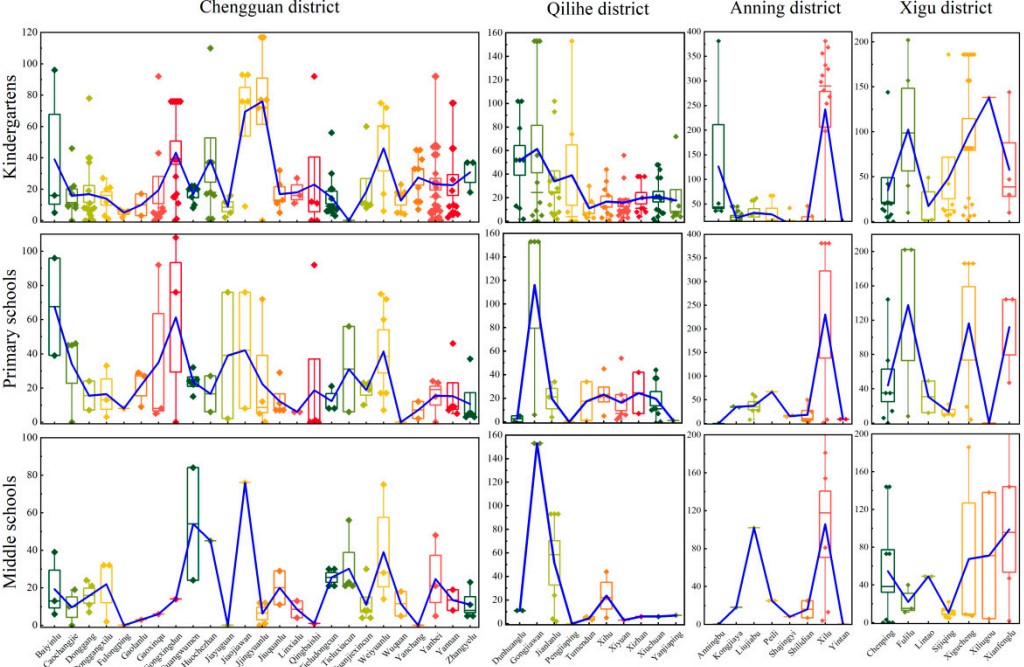

**Figure 5.** Differences in the supply capacity of basic educational facilities. (Different colors represent different streets. For greater clarity, we have marked the average values of all indicators with blue lines.).

### 3.3. Accessibility of Basic Educational Facilities

In the UNA tool, each residential building was set as the starting point, and the facility was set as the end point. We imported road network data and searched around each residential building within 300 m of kindergartens, 500 m of primary schools, and 800 m of middle schools. The number of basic educational facilities that each residential building can reach under walking conditions is the following: the accessibility is three, indicating that the residential building can reach three basic educational facilities of this type at the same time.

As shown in Figure 6, the spatial pattern of accessibility of basic educational facilities indicates that there are significant differences in accessibility between both different and the same types of facilities. The accessibility of basic educational facilities was high in the northwest and southeast corners of the urban area of Lanzhou. There were significant differences in accessibility on both sides of urban roads in residential areas. Of residential buildings, 34.83% could reach kindergartens, 27.43% of residential buildings could reach primary schools, and 21.11% of residential buildings could reach middle schools. Thus, kindergartens had the highest accessibility, followed by primary schools, and middle schools had the lowest accessibility. From the perspective of accessibility, the highest accessibility of kindergartens was 21, accounting for 8.87% of the total accessibility. An accessibility value of one was found the most, accounting for 30.69%. There were 5335 residential buildings with primary school accessibility values higher than 1 (accessibility values were 2, 3, 5, 7, 10), accounting for 69.31%. The highest accessibility to middle schools value was 11, and the number of residential buildings with an accessibility value of 1 was 1444, accounting for 30.95% of buildings.

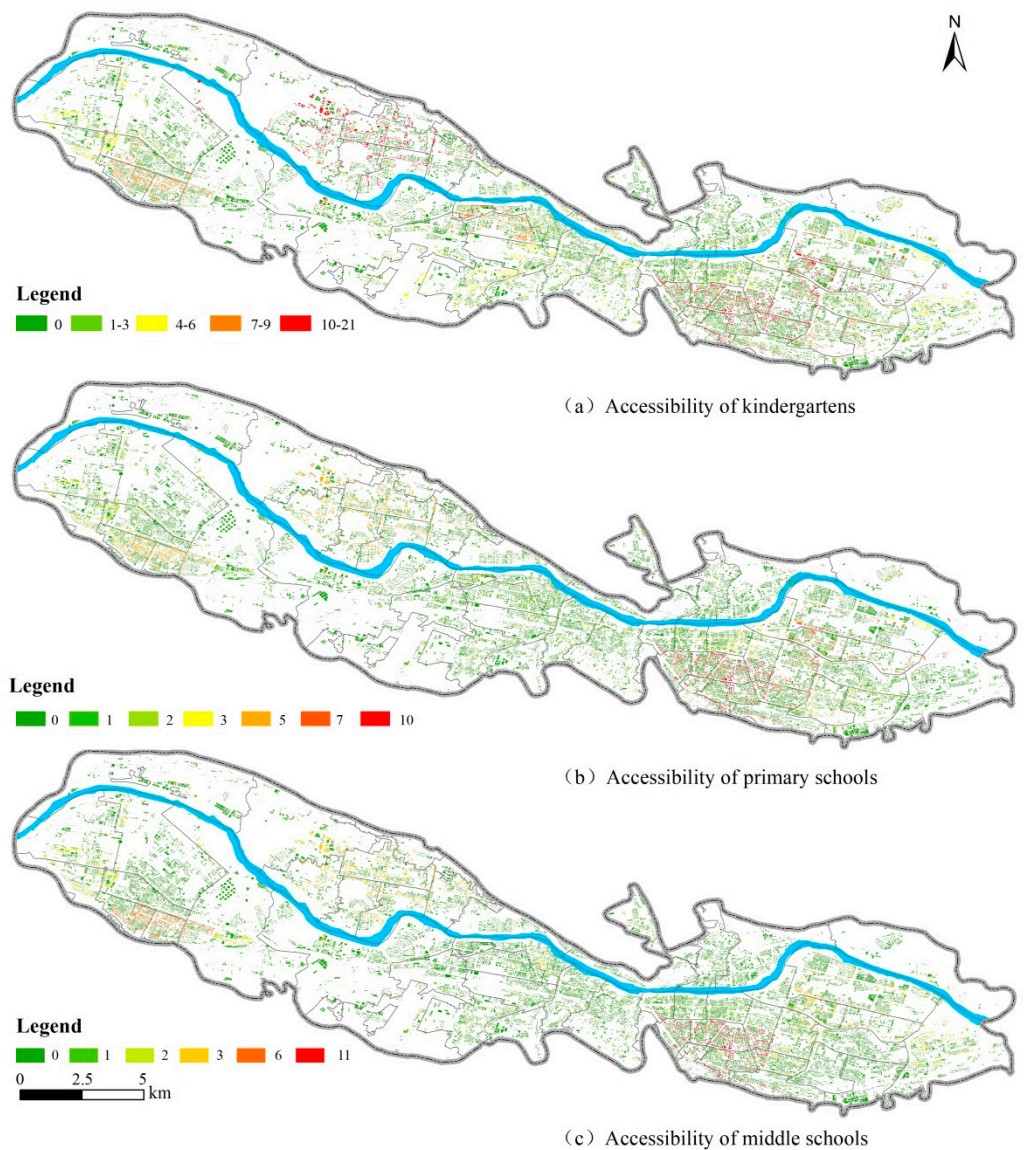

**Figure 6.** Accessibility of basic educational facilities.

The number of different types of educational facilities accessible in the urban area of Lanzhou City varied greatly. There were a large number of residential buildings where the

required educational facilities could not be reached under walking conditions. In addition, there were differences in the quantity of demand in each district. There were also significant differences in the proportions of demand in the districts. The demand for kindergartens in each district was balanced. There were 6835 buildings in Chengguan District lacking walkable access to kindergartens, accounting for 63.49% of the total residential buildings. There were 1723 such buildings in Anning District, accounting for 65.89% of the total residential buildings. There were 3154 such buildings in Qilihe District, accounting for 63.95% of the total residential buildings. The number of buildings in Xigu District accounted for the largest demand, with 65.97% of the total residential buildings lacking walkable access to kindergartens. There were significant differences in the demand for primary schools in different districts. The demand in Chengguan District and Qilihe District was significantly higher than that in both Anning District and Xigu District. Buildings lacking walkable access accounted for 74.83%, 74.77%, 68.01%, and 67.15% of the total residential buildings in the district, respectively. The demand for middle schools in Qilihe District was the greatest. A total of 4254 residential buildings could not reach middle schools on foot, accounting for 86.25% of the residential buildings in the district. The demand for middle school in Anning district was the least. A total of 1964 residential buildings were unreachable, accounting for 72.36%.

In Figure 7, comparing the basic educational facilities, it can be seen that the demand for middle schools in each district was greater than that of both primary schools and kindergartens. The number of middle schools was much smaller than that of kindergartens. The existence of a corresponding increase in walking distance was the main reason for the significant difference in demand. Each street had the same demand for basic educational resources at all levels. However, there were certain differences in the specific demand for basic educational facilities at all levels. Donggang, Donggang West Road, Guangwumen, Jingyuan Road, Jiuquan Road, Railway West Village, Yanbei Street in Chengguan District, Shajingyi and West Road Streets in Anning District, Yanjiaping Street in Qilihe, and Xianfeng Road Street in Xigu District had the greatest demand for educational facilities. However, the demand for kindergartens on Chenping, Peili, and Xiliugou Streets was greater than that for other basic educational facilities. The government should pay attention to the demand for educational facilities in districts at the macro level, and plan and allocate educational facilities according to the differences in demand for educational facilities at all levels, and on each street.

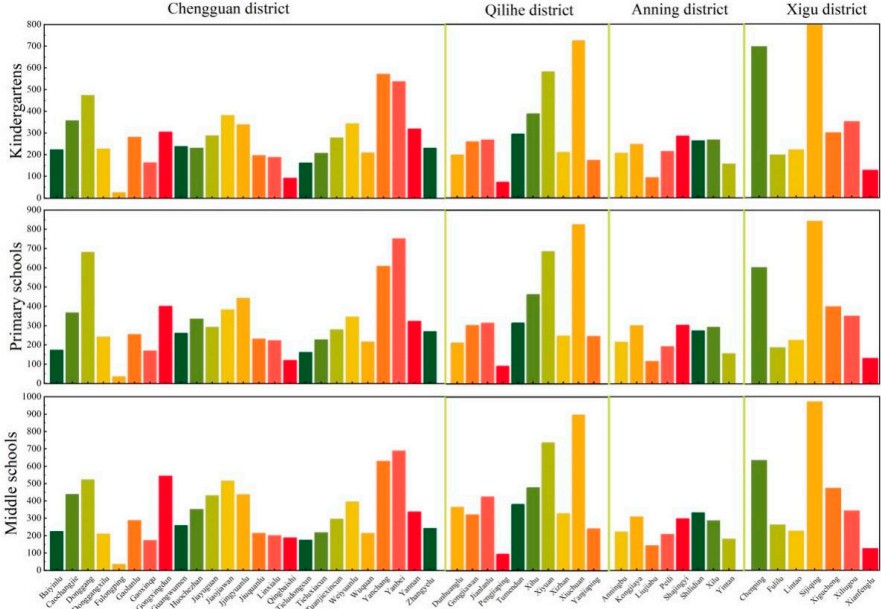

**Figure 7.** Differences in demand for basic educational facilities. (Different colors represent different streets).

## 4. Discussions

The supply and demand of basic educational facilities in the urban area of Lanzhou showed significant differences at the inter-district level, at the street level, and even within individual residential building areas. External conditions and internal factors, such as urban spatial structure, traffic road network structure, and social spatial behavior can affect the spatial equity of basic educational facilities. We first summarized the influence of external and internal factors, then put forward a targeted optimization configuration plan, and pointed out the limitations of this study.

### 4.1. Factors for the Distribution of Basic Educational Facilities

#### 4.1.1. The Historical Development

The history of urban development and changes in Lanzhou strongly affected the distribution of basic educational facilities [37]. Chengguan District developed in the early stage, and had a large number and dense distribution of basic educational facilities. However, due to the dense population distribution and agglomerated residential buildings, the basic educational facilities in this area had a difficult time meeting the demand. The development of Xigu District lagged behind that of Chengguan District. However, Xigu District benefited from the planned economic policy in the early days of the founding of the People's Republic of China, when basic educational facilities were organized around large enterprises. The continuous production and housing-supporting model formed the basic distribution of basic educational facilities [38]. This affected the spatial equity of interregional basic education for a period of time. As the main body of the education service supply, the government's policy played an important role in the development of interregional differences. Anning District is known as the "University City", and is an important education sector. The basic educational supply capacity of Anning District was significantly higher than that of other districts.

#### 4.1.2. Physical and Geographical Factors

Physical geography is the basis for the extension of urban development. Lanzhou is restricted by its unique valley basin of "two mountains and one river". This is consistent with the spatial structure of the downtown area of Lanzhou. The beaded urban outline developed on river terraces, the fragile ecological environment, limited space development potential, and mountain barriers caused the extension direction of the urban road network. Those limited the flow of information and resource conversion between streets [39]. The long east–west distance and narrow north–south distance of Lanzhou affected the streets' extent and outline. This affected the range of basic education services and the allocation of basic educational facilities at the street level. In addition, we utilized a top-down elite planning method and a planning management model based on administrative units [40]. Streets are adjacent but cannot share basic education services, resulting in differences in the supply and demand of various basic educational facilities at the street level.

#### 4.1.3. Socioeconomic Factors

Achieving a livable socioeconomic level is the basic guarantee offered to citizens, and understanding it provides a material basis for matching the demand and supply of basic educational facilities. The improvement in the level of economic development has made funding sources for the construction of basic educational facilities more diversified, and the funding guarantee for basic educational facilities has become more stable. Due to the differences in the development of the urban areas of Lanzhou City, social and economic development varied greatly among the areas. The better the economic level, the greater the investment in basic educational facilities. A high economic level can lay a solid material foundation for the construction of basic educational facilities. In addition, as the demand of people for a better quality of life is increasing day by day, the demand for basic educational facilities was characterized by being for high-quality facilities. Residents' willingness to receive a basic education has become stronger, thereby improving the distribution of basic

educational facilities. In addition, the population is the main body of basic education public services. The spatial distribution of basic educational facilities should take into account the regional population density. Therefore, the quantity and type of basic educational facilities can adapt to the degree of regional population agglomeration.

### 4.1.4. Travel Factors of Residents

Under a travel mode, or a certain travel distance, the stronger the supply capacity of basic educational facilities, the better the accessibility from residential buildings. Affected by spatial distance, traffic conditions, and traffic modes, the interiors of the residential areas showed hierarchical differences from the inside to the outside, that is, in the accessibility of "home–school" [41]. There are inequities in basic educational resources that groups at different living locations can obtain. Vulnerable groups in cities, such as people living in urban villages, have a very low degree of sharing of high-quality educational resources. There is room for improvement in the spatial equity of basic educational facilities.

### 4.2. *Optimal Spatial Allocation for Basic Educational Facilities*

Education can facilitate bottom-up mobility in society. The spatial inequity of basic educational facilities increases the spatial differentiation of the resident population, forces different groups of people to increase the effectiveness of their basic education, inhibits the flow of social classes, and hinders social progress [42]. The government must insist on "equity in education space" and solve the problem of spatial imbalance through multiple plans.

### 4.2.1. Implementing Refined Planning Management and Resource Allocation

According to the requirements of urban stock development, the government should effectively connect with the urban renewal system, ensure the flexibility and comprehensive effects of the supply of basic educational facilities, and implement refined planning management and resource allocation [43]. The government guarantees the space supply of basic educational facilities through non-land expansion methods, including building renovation and land use compatibility efforts. In addition, the government should make full use of the vacated land in urban areas to solve the problem of the lack of basic educational facilities in surrounding communities. According to the needs for a given school's construction and for an increase in educational capacity, the government should cooperate with the planning department to implement the construction of the school, and the renewal and development at the same time, giving priority to the use of vacated land, and even build it in advance, based on demand. In addition, the accessibility of roads should be strengthened. The government should increase the density of the walking network and pay attention to its replacement by other modes of transportation. The government should guide parents of students in kindergartens, as well as more senior students, to reasonably use buses, bicycles, and shared bicycles. Multi-transportation modes can alleviate the spatial inequalities of basic educational facilities between districts and streets.

### 4.2.2. "Living Circle of Community": Promoting the Balanced Development of Basic Education

The traditional matching model used the "thousand people index" as the control index. However, living circle planning relies on a change of planning and thinking modes. Living circle planning opens up new planning ideas for the construction of livable cities, in terms of achieving uniformity and a grass-roots level of public services. The government should determine school sizes based on the new idea of building a "Living Circle of Community", with regard to the scale of the living circle and the demand for educational facilities. In addition, the government should control the scale of residential areas at different levels, and improve the discrepancy between supply and demand when it appears in the internal spaces of oversized residential areas. To ensure educational equity and achieve equal education in the same city, the government should expand the supply of some high-quality

educational resources, in the form of branch schools, to improve the quality of surrounding education, share the burden of high enrollment pressure, and realize the relocation and integration of basic educational resources.

### 4.2.3. Combining Big Data to Establish an Early Warning Mechanism of Academic Degree

There is a certain cyclic difference that emerges between the supply of construction space and the construction of schools in urban planning, that is, the educational position of students increases year by year. There is a certain dislocation between the demand for educational achievements of students and urban space preservation. Therefore, the supply and demand of an academic degree can fail to achieve a dynamic balance, thereby increasing teaching pressure. This study suggests that relevant education departments should establish a cooperative relationship with big data platforms, use big data information and new technologies, and cooperate with public security, health, and other departments to collect data information, such as household registration population, birth population, and the total number of existing degrees. The government should dynamically monitor and analyze the distribution of school-age children based on factors such as changes in mobility, and coordinate efforts with the demand for academic degrees. Early warning can be given at different levels in places where there is a shortage of academic degrees being granted. Parents' expectations can be reasonably guided, the planning and construction of schools can be accelerated, and the distribution of basic educational facilities and academic degrees in school can be adjusted in time.

### 4.3. Limitations and Further Study

Based on spatial accessibility, this paper measured the equity of basic educational facilities and regional differences in supply and demand, and optimized the spatial distribution of educational facilities. The method described in this paper can serve as a reference for evaluating the spatial equity of other public service facilities in the city. However, there are some limitations in this paper. First, this study allocated students equally to each residential building, ignoring the differences between regions and groups of people. There were certain deviations in some regions. Second, we only took into account the supply and demand of basic educational facilities under walking conditions. The various modes of transportation in the city shorten the spatial distance, causing changes in accessibility, and have a certain impact on the actual supply and demand of basic educational resources. Third, the equity of educational resources affects the academic performance of students in schools to some extent, and how the equity of educational resources ultimately affects academic performance is not discussed in this paper. Therefore, in future research, more research should be done on residents' travel behaviors, guided by the demands of different regions and groups of residents, thereby providing a more accurate recommendation for the allocation of basic educational facilities. Attention should also be paid to the exploration of the causes and effects of the equity of educational resources.

### 5. Conclusions

This paper evaluated the regional equity of basic educational facilities in Lanzhou City from the perspective of supply and demand through urban network analysis tools. The results are as follows. All kinds of basic educational facilities have significant river tendencies. There were a lot of dislocations, with a small set of spatial overlaps between facilities and residential buildings. From the perspective of walking conditions, the supply capacity of various basic educational facilities was weak. Students in the school district need to use certain means of transportation or take a longer time. There were large differences in the supply capacity among urban areas, and in supply levels in the inner streets of each district. Under the walking condition, the accessibility of kindergartens was the best at 34.83%, followed by primary schools at 27.43%, and middle schools were the lowest at 21.11%. The demand for various types of educational facilities in Chengguan District and Qilihe District was significantly higher than that in Anning District and Xigu District. Each

street had the same demands for types of basic educational resources at all levels, but there were differences in the demand with regard to quantity.

**Author Contributions:** Conceptualization, Q.H. and X.C.; Writing—Review & Editing, Q.H. and L.M.; Funding acquisition, Q.H.; Investigation, Q.H., X.C. and L.M.; Methodology, X.C.; Writing—Original Draft, X.C.; Visualization, X.C. All authors have read and agreed to the published version of the manuscript.

**Funding:** This research was funded by the Soft science research project of Gansu Province, grant number 21CX6ZA103; Gansu Social Science Planning Youth Project, grant number 2022QN038.

**Informed Consent Statement:** Informed consent was obtained from all subjects involved in the study.

**Data Availability Statement:** Data involving personal personal information is inconvenient to disclose.

**Conflicts of Interest:** The authors declare no conflict of interest.

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
