# Peer review of "The Equity of Basic Educational Facilities from the Perspective of Space"

_sustainability, doi:10.3390/su151512031_

Round 1

Reviewer 1 Report

The topic is interesting and will have a good attraction to readers.  But it has some shortcomings also. 
  1. Structure the Abstract to make it more clear. For example, The abstract is very lengthy and unstructured. Even if it is unstructured it must have objective, methodology, results, conclusion, implications. 
  2. The literature review is short and has no references. This is evident from the introduction. It is strongly recommended to add references as suggested at least 50.
  3. I believe that the methodology and analysis is reasonable and satisfactory.
  4. The conclusion is well written.
  5. I recommend the paper for publication after these changes have been made.
  6. Referencing style within the text is wrong. Please change it according to the journal format
  7. Add references. At least 50 references are required for reasonable review.
  8. Please focus on latest references

Reviewer 2 Report

The paper is of interest only for the local education policy makers. A comparison with other countries/regions would improve the paper's value.

A deeper discussion on the influence of distribution of education resources on students' academic results would be welcome.

It is an interesting topic, but the results can hardly be generalized.

There are not many articles on the subject, it could be a source of inspiration for other authors. I feel that the relevance is low as it regards only a part of China.

There are some mistakes regarding word order and subject-verb agreement.

Reviewer 3 Report

The topic "The Equity of Basic Educational Facilities from the Perspective of Space" is interesting and has a good research significance, and the author has chosen Lanzhou as the study area, which is representative. The results of the study can be used as a reference for other similar cities. Revisions:

1. The author only briefly discusses the background of the study, but lacks the necessary literature discussion, and suggests adding literature discussion on the important research aspects of this thesis.

2. In terms of literature citation, most of them are Chinese scholars at present, and it is suggested to increase the research of foreign scholars. Especially in the literature discussion and "4. Discussion (P12-14)" section.

3. The part of "5. Conclusion" is too brief, and the core ideas of the paper can be further condensed and added appropriately.

 Minor editing of English language required.

Reviewer 4 Report

1. References are missing in many parts, including but not limited to Sections 2.1, 2.3, etc.

2. Section 2.1. Overview of the study area: 

(1) There's redundant introduction about Lanzhou in the first and second paragraphs. 

(2) End of second paragraph: it's better to include what the national standards are, for the average school area per student and average sport area per student.

3. What's the reason for choosing Chengguan District, Qilihe District, Anning District and Xigu area as examples? Based on Figure 1, geographically they are next to each other, but readers may also be interested to see on Figure 1 about where the approximate location they are in Lanzhou.

4. It would be better if there is more explanation about how to interpret the distribution of kernel density which is shown in many figures: the legends only say Low and High and 0-2.5-5 km.

5. What are the blue lines in Figures 2, 3, and 4?

6. Figure 5: are the curving lines connecting the mean values for each school? It would be better if another color could be used for the lines, for example blue, so that it would stand out more clearly and not mixed with the color of many of the columns.

Round 2

Reviewer 2 Report

Dear authors, thank you for taking into consideration my suggestions. I understand now, after reading your comments, that this study could be seen as an example of good practice as well as inform us of the realities present in one part of the world and serve as basis for comparison.

Author Response

We thank the reviewers for recognizing our work. And thank you very much for your hard work.

Reviewer 3 Report

After the revision of the thesis, the literature part is much better, and the discussion is clearer, smoother, easier to read, and logically clearer. This paper is an interesting study, and the research conclusions can provide a basis for related research. It is recommended that the English part be further polished and accepted for publication.

 Minor editing of English language required

Author Response

(The authors gave the same response as above.)

Reviewer 4 Report

My previous concerns/comments have all been appropriately addressed.

Author Response

(The authors gave the same response as above.)
